# Removing Calcium Ions from Remelt Syrup with Rosin-Based Macroporous Cationic Resin

**DOI:** 10.3390/polym14122397

**Published:** 2022-06-14

**Authors:** Gege Cheng, Wenwen Li, Long Li, Fuhou Lei, Xiuyu Liu, Qin Huang

**Affiliations:** 1School of Chemistry and Chemical Engineering, Guangxi Minzu University, Nanning 530006, China; ggcheng2022@163.com (G.C.); l19981207654321@163.com (W.L.); lilong19980227@163.com (L.L.); leifuhougxun@126.com (F.L.); xiuyu.liu@gxun.edu.cn (X.L.); 2Key Laboratory of Chemistry and Engineering of Forest Products, State Ethnic Affairs Commission, Nanning 530006, China; 3Guangxi Key Laboratory of Chemistry and Engineering of Forest Products, Nanning 530006, China; 4Guangxi Collaborative Innovation Center for Chemistry and Engineering of Forest Products, Guangxi Minzu University, Nanning 530006, China

**Keywords:** rosin-based macroporous cationic resin, calcium ions, remelt syrup

## Abstract

Mineral ions (mainly calcium ions) from sugarcane juice can be trapped inside the heating tubes of evaporators and vacuum boiling pans, and calcium ions are precipitated. Consequently, sugar productivity and yield are negatively affected. Calcium ions can be removed from sugarcane juice using adsorption. This paper described the experimental condition for the batch adsorption performance of rosin-based macroporous cationic resins (RMCRs) for calcium ions. The kinetics of adsorption was defined by the pseudo-first-order model, and the isotherms of calcium ions followed the Freundlich isotherm model. The maximal monolayer adsorption capacity of calcium ions was 37.05 mg·g^−1^ at a resin dosage of 4 g·L^−1^, pH of 7.0, temperature of 75 °C, and contact time of 10 h. It appeared that the adsorption was spontaneous and endothermic based on the thermodynamic parameters. The removal rate of calcium ions in remelt syrup by RMCRs was 90.71%. Calcium ions were effectively removed from loaded RMCRs by 0.1 mol·L^−1^ of HCl, and the RMCRs could be recycled. The dynamic saturated adsorption capacity of RMCRs for calcium ions in remelt syrup was 37.90 mg·g^−1^. These results suggest that RMCRs are inexpensive and efficient adsorbents and have potential applications for removing calcium ions in remelt syrup.

## 1. Introduction

Desalination of carbonated and filtered remelt syrup is an indispensable step in the sugar industry. The inorganic minerals include calcium ions (mainly), magnesium ions, iron ions, silicic acid, phosphate, and carbonate ions in sugar juice [1]. More than 90% of the inorganic matter in the syrup relates to calcium [2]. As the concentration of sugar juice increases in the evaporators, calcium ions are precipitated as their solubilities are exceeded [3], which directly affects the productivity and achievable yield of sugar [4]. Helmut et al. [1] claimed that calcium ions in a beet juice evaporator could be reduced by 80–90% using KEBO DS (a scale inhibitor). Typically, anionic polymers, such as polyacrylics and poly (amino polyether tetra-methylene phosphonic acid), are used in the sugar industry to inhibit calcium salts [1]. The use of scale inhibitors in the sugar industry is also limited by health concerns. These inhibitors need to be approved by relevant agencies (for example, the United States Food and Drug Administration) before they can be used in the sugar process [1]. The desalination process has been explored by many eco-friendly and green technologies, including membrane separation [5], ozonation [6], coagulation [3], and adsorption [7]. Researchers prefer adsorption because it is inexpensive, widely available, and easy to use; additionally, it has the potential to handle large-scale production [8]. The preparation and functionalization of new decalcification adsorbents for remelt syrup with high adsorption performance, environmental friendliness, and low cost has become the research focuses.

The macroporous cationic resins (MCRs) used in the syrup are mainly synthesized based on styrene and divinylbenzene [9]. Styrene and divinylbenzene have been classified as class 2B carcinogens, which are limited by health concerns, by the International Agency for Research on Cancer of the World Health Organization (IARCWHO) [10]. With the improvement of people’s living standards, requirements for food quality and safety also increase. Thus, it requires us to investigate green biomass-based decalcification adsorbents that are highly adsorption-efficient, low-cost, and recyclable.

About 90% of crude rosin is rosin acid, which is derived from the exudation of conifer trees [11]. Resin acid (rosin) can be modified by an esterification and (or) addition reaction based on the conjugated double bonds and carboxyl group [12]. In the prior study, we successfully prepared a novel ethylenediamine rosin-based resin (EDAR) for the removal of phenolic compounds from water. The interaction model and adsorption mechanism of EDAR-adsorbed phenolic compounds in water were studied, which provided the basis for its application [13]. Similarly, Li et al. prepared a new rosin-based sugarcane juice decolorization agent, where the crosslinking agent was modified rosin, silica was the carrier, and the quaternary ammonium cation was the functional group [8]. The excellent mechanical properties and thermal stability of the adsorbent are due to the specific three-membered phenanthrene ring structure of rosin [14]. Thus, modified rosin has the potential to produce green, economical, and eco-friendly adsorbents in the sugar industry. However, as far as we know, rosin-based resins have not been prepared and used for removing calcium ions from remelt syrup.

In this study, rosin-based macroporous cationic resins (RMCRs) were prepared using modified rosin ethylene glycol maleic rosinate acrylate (EGMRA) as a cross-linking skeleton. The potential use of the RMCRs in removing calcium ions from remelt syrup was described. Furthermore, the mechanisms of the RMCRs adsorption of the calcium ions were elucidated by using an adsorptive isotherm and kinetic models and calculating thermodynamic parameters.

## 2. Materials and Methods

### 2.1. Materials

Hydrochloric acid (HCl, 37–39%), calcium chloride (AR), and sodium hydroxide (AR) were purchased from Sinopharm Chemical Reagent Co., Ltd. (Shanghai, China). EGMRA was provided by the Guangxi Key Laboratory of Forest Products Chemistry and Engineering (Nanning, China) [15]. The remelt syrup after carbonation and decoloration by anion resin was kindly provided by Fangcheng Sugar Refinery (Fangchenggang, China). Commercial resins, including four derivatives of the styrene-divinylbenzene copolymer (types: FPA51, FPC22 Na, FPA40 Cl, FPA90 Cl, FPC14 Na, FPC23 H, and FPC22 H), and one polymethacrylic acid (type: FPA98 Cl) were purchased from Dow Chemical (Shanghai, China).

### 2.2. Preparation and Characterization of RMCRS

The RMCRs used in this study were all self-prepared, and followed a previous preparation process with some modifications [16]. The functional monomers MAA (6.32 g), the porogen polypropylene glycol (1.39 g), the cross-linker EGMEA (20.25 g), and AIBN (0.2 g) were dissolved in ethyl acetate (60 mL) by sonication to obtain an organic phase. SDS (0.02 g) and PVA (0.02 g) were dissolved in deionized water in a 250 mL three-necked flask, and then the organic phase was added at 60 °C. The mixture was thermally polymerized at 80 °C for 8 h and stirred at 200 rpm. The resins were extracted with ethanol and deionized water, then immersed in 3.0% NaOH solution to ionize the COOH groups to COO− groups, and ultimately washed continuously with deionized water until the pH was approximately 7.0.

### 2.3. Static Adsorption and Regeneration of the Resins

The optimum adsorbent dosage, temperature, pH value, and time for calcium ions adsorption by RMCRs were determined by preliminary experiments. The adsorption of calcium ions onto the RMCRs was carried out in a conical bottle containing 50 mL of calcium ions solution and 0.200 g of RMCRs. The adsorption isotherms of calcium ions with different initial concentrations (30, 60, 90, 120, and 150 mg·L^−1^) were obtained at temperatures of 328, 338, and 348 K. Kinetic experiments were conducted in glass flasks that contained 500 mL of calcium ions solutions at a pH of 7.0 with a calcium ions’ initial concentration of 150 mg·L^−1^ and 2 g of RMCRs. The conical bottles were oscillated at 100 rpm at 348 K and sampled at regular time intervals. The concentration was determined by ICP-OES (iCAP 600 Seris, Thermo Fisher Scientific, Waltham, MA, USA). The pH value was adjusted by HCl and NaOH solutions with concentrations of 0.1 mol·L^−1^, and the effect of pH value on calcium ion adsorption was investigated. The resins after adsorption were shaken with 0.1 mol·L^−1^ of HCl for 12 h at 298 K for regeneration. RMCRs were tested for reusability through adsorption regeneration cycles.

### 2.4. Fixed-Bed Column Experiments

The fixed-bed on calcium ions adsorption was conducted in silica sand glass columns (Ø1.5 × 20 cm^2^) filled to depths of 4 cm with RMCRs. Under the drive of the pressure provided by a peristaltic pump, the syrup (45 °Bx, pH = 7.0) was passed through the column at a rate of 2.0 mL·min^−1^, and the effusive calcium ions solutions were determined at various intervals.

### 2.5. Analysis

The RMCRs before and after the adsorption of calcium ions were characterized by Fourier-transform infrared spectroscopy (FTIR) (Nicolet 5700, Thermo Fisher Scientific, Waltham, MA, USA), X-ray diffraction (XRD) (Siemens D5000 diffractometer, Bruker, Germany), X-ray photoelectron spectroscopy (XPS) (ESCALAB 250Xi, Thermo Fisher Scientific, Waltham, MA, USA), and field emission scanning electron microscopy (FE-SEM) (JSM-7500F, JEOL, Tokyo, Japan) with energy-dispersive spectrometry (EDS). The zeta potentials of RMCRs were measured with a Zeta Potential Analyzer (Zetasizer 2000 Analyzer, Malvern, UK) at an initial pH ranging from 2.0 to 12.0.

After each experiment, the solution was filtered through 0.45 µm filters and the concentration of calcium ions was analyzed by inductively coupled plasma–atomic emission spectroscopy. All the adsorption/regeneration experiments were performed at 100 rpm with triplicates and the results were averaged from all replicates. The adsorption efficiency (*q_t_*) and removal rate (*R*) were calculated as follows:(1)$$q_{t}=\frac{\left(C_{0}-C_{t}\right)\times V}{m}$$
(2)$$R\left(\%\right)=\frac{C_{0}-C_{t}}{C_{0}}\times 100$$
where *q_t_* represents the amount (mg·g^−1^) of calcium ions adsorbed at time *t* (min); *C*_0_ (mg·L^−1^) and *C_t_* (mg·L^−1^) are the initial concentration and *t* (min) concentration of calcium ions, respectively; *V* (L) is the volume of the calcium ions solution; and *W* is the weight of the RMCRs.

## 3. Results and Discussion

### 3.1. Characterization of the RMCRs

#### 3.1.1. N_2_ Adsorption–Desorption Isotherm Analysis

The pore structures and specific surface areas of the RMCRs were determined by using adsorption–desorption experiments, as shown in Figure 1a. The RMCRs’ isothermal curves also showed type II curves with well-defined H_1_ hysteresis-type loops, thus inferring cylindrical pores with a uniform macroporous structure [17]. The pore structures of the RMCRs had an average pore diameter of 42.40 nm (Figure 1b), Brunauer–Emmett–Teller (BET) surface area of 10.24 m^2^·g^−1^, and cumulative pore volume of 22.20 mm^3^·g^−1^. Thus, the RMCRs could exhibit excellent adsorption abilities on account of their high interconnect pores and high permeability, which facilitates the diffusion of adsorbents.

#### 3.1.2. TGA Analysis

The thermal stability of the RMCRs was characterized by thermogravimetric analysis (STA449F3, Netzsch-Gerätebau GmbH, Selb, Germany). The results are presented in Figure 1c. RMCRs began to decompose at 220 °C due to the decomposition of rosin [18]. The onset temperature for RMCRs decomposition was approximately 220 [19]. The apparent weight loss of RMCR mainly occurred at temperatures ranging from 300 to 450 °C. The temperature of desalination in the sugar industry is commonly under 100 °C. Therefore, the RMCRs have high thermal and chemical stability and are appropriate for removing calcium ions from remelt syrup.

#### 3.1.3. FE-SEM Analyses

The morphology and size of RMCRs were characterized by FE-SEM, and representative images are shown in Figure 1d,e. In the panoramic image, RMCRs were regular spheres with smooth, porous surfaces. The internal holes of the RMCR were interconnected. The rich porous structures of the RMCRs not only promoted the liquid mass transfer but also better access to the interaction sites [20]. Thus, it is beneficial to elevate the adsorption of calcium ions from remelt syrup [14].

### 3.2. Static Adsorption Experiments

#### 3.2.1. Effect of RMCRs Dosage

The influences of RMCRs dosage on the adsorption capacity of calcium ions are presented in Figure 2a. With the increase in RMCRs dosage, the effective adsorption area increased, hence the removal efficiency promotion. However, the increase in the resin dosage increased the unsaturated loca on the adsorbent surface, thereby decreasing adsorption efficiency. In all the subsequent experiments, the solid-to-liquid ratio of 4.0 g·L^−1^ was selected as the optimal RMCRs dosage for calcium ions adsorption in consideration of efficiency and economy. Hence, RMCRs have great potential use in the removal of calcium ions from remelt syrup.

#### 3.2.2. Effect of Temperature

As scanned in Figure 2b, the temperature rose from 308 K to 348 K, and the *q_e_* increased from 27.4 mg·g^−1^ to 37.5 mg·g^−1^ under other equal conditions. The increase in temperature enhanced the binding of calcium ions to RMCR adsorption sites, due to the movement of calcium ions in the solution being accelerated. Hence, 348 K was selected as the optimal temperature for calcium ions adsorption. The adsorption of calcium ions by RMCRs is an endothermic process.

#### 3.2.3. Effect of Contact Time

According to Figure 2c, the calcium ions absorption capacity over RMCRs gradually increased in the beginning, and then steadily reached equilibrium after 600 min, which indicates a large number of active sites on the surface of the adsorbent in the initial stage. However, the adsorption sites were occupied, and the adsorption rate slowly decreased until equilibrium was reached. Adsorption equilibrium was reached in approximately 10 h, the *q_e_* of RMCR was 37.05 mg·g^−1^, and the corresponding removal rate was 90%. Thus, the contact time was set to 10 h in subsequent experiments.

#### 3.2.4. Effect of pH

The pH plays a vital role in the adsorption of calcium ions by RMCRs. The effects of initial pH (2.0–8.0) on the calcium ions adsorption by the RMCRs are shown in Figure 2d. The zeta potential value was correlated with the charge of the RMCRs and reflected their adsorption characteristics. Here, we focused on the main functional group of RMCRs involved in calcium ions adsorption, namely, the −COONa groups. This group could be very easily ionized to form −COO− groups. The zeta potentials of RMCRs are shown in Figure 3, with pH_pzc_ values of 2.7. At pH = 2 (pH < pH_pzc_), the removal rate of calcium ions was lower, due to the positive charge on the surface of RMCRs; thus, the electrostatic repulsion interrupted the adsorption of calcium ions. When the solution pH = 3 (pH > pH_pzc_), the surfaces of the RMCRs acquired negative charges. Thus, calcium ions and resins are attracted by electrostatic forces. Therefore, in the range of pH 2.0 to 3.0, the removal percentages sharply rose from 60.87% to 84.08%. When the pH > 4.0, the adsorption capacity was basically constant, as confirmed by the pH_pzc_ (zero charge points) and zeta potential of RMCRs. At pH 8.0, the adsorption capacity of RMCRs decreased because the system could present molecular agglomeration due to the number of intermolecular interactions and induced precipitation processes or an increase in viscosity [21]. The supreme adsorptivity for calcium ions was obtained at pH 7.0. Therefore, pH 7.0 is considered the optimum condition and used hereafter.

### 3.3. Adsorption Kinetics

The absorption data were investigated by the pseudo-first-order kinetic, pseudo-second-order kinetic, and intraparticle diffusion equations to explore the calcium ions capacity of RMCRs. Linear forms of equations of these models can be written as follows:

Pseudo-first-order kinetic equation [22]:(3)$$\log\left(q_{e}-q_{t}\right)=\log q_{e}-\frac{K_{1}}{2.303}\cdot t$$

Pseudo-second-order kinetic equation [23]:(4)$$\frac{t}{q_{t}}=\frac{1}{K_{2}\cdot q_{e}^{2}}+\frac{t}{q_{e}}$$

Intraparticle diffusion equation [8]:(5)$$q_{t}=K_{3}\cdot t^{0.5}+C$$

In the above equations, *q_t_* and *q_e_* are the calcium ions adsorption capacity (mg·g^−1^) at time *t* and equilibrium time, respectively; *K*_1_ (min^−1^), *K*_2_ (g·mg^−1^·min^−1^), and *K*_3_ (mg·g^−1^·min^0.5^) are the rate constants of the pseudo-first-order kinetic, pseudo-second-order kinetic, and intraparticle diffusion equations, respectively; *t*
^0.5^ (min^0.5^) is the square root of the contact time; and *C* represents the boundary layer thickness.

The fitting results are shown in Figure 4a–c, respectively. The pseudo-first-order equation explains the experimental data well compared with the other two equations, and the adsorbed quantities calculated (35.60 mg·g^−1^) by this model are closer to those determined experimentally (37.05 mg·g^−1^); thus, the main rate-limiting step of adsorption is physical adsorption. However, Figure 4c illustrates the separation of the adsorption process into two stages, namely fast adsorption and slow adsorption. Initially, calcium ions are adsorbed onto the surfaces of the RMCRs; after the surfaces are saturated, they gradually filter into the pore and inner surfaces of RMCRs by intraparticle diffusion until sorption decreases to equilibrium. Therefore, the calcium ions in aqueous solutions adsorbed onto RMCRs is a complicated procedure that involves boundary layer and intraparticle diffusion.

### 3.4. Adsorption Isotherms and Thermodynamics

Adsorption isotherms facilitate the description of the interaction between calcium ions and RMCRs’ surfaces at equilibrium. The adsorption isotherms at 328, 338, and 348 K are shown in Figure 5. The *q_e_* of RMCRs increased with the concentration of calcium ions and initial temperature.

The isotherm data were fitted by the Freundlich and Langmuir models to investigate the adsorption behavior. The experimental data of isotherm models are generally employed to delineate by adsorption equations and can be written as follows:

Freundlich model [22]:(6)$$q_{e}=k_{F}\cdot C_{e}^{\frac{1}{n}}$$

Langmuir model [24]:(7)$$q_{e}=\frac{q_{m}\cdot k_{L}\cdot C_{e}}{1+k_{L}\cdot C_{e}}$$
where *K_F_* (mg·L^−1^) represents the Freundlich constant; 1/*n* is the strength of adsorption; and *K_L_* (L·mg^−1^) and *q_m_* (mg·g^−1^) are constants correlated with the affinity of the adsorption sites for RMCRs, respectively.

Table 1 summarizes the fitting results, in which the Freundlich and Langmuir models depict the isotherm data adequately. *N* is related to the adsorption driving force and the energy distribution of the adsorption sites. As shown in Table 1, the values of 1/*n* are between 0.1 and 0.5, which indicate that the adsorption of calcium ions on RMCRs is facile. As a result, the Freundlich model performed better (*R*^2^ > 0.99) for describing the adsorption system in the range of concentrations and temperature ranges studied. Hence, calcium ions are adsorbed as a heterogeneous surface of an adsorbent and multilayer. Some heterogeneities on the surface of the RMCRs will take effect in calcium ions adsorption because of the existence of carboxyl and the abundant pore structures of RMCRs. These results further demonstrate the excellent promise in the removal of calcium ions by RMCRs in the sugar industry.

At 348 K, the adsorption capacity of calcium ions increases, illustrating the endothermic nature of the adsorption process.

Equations (8) and (9) were applied to the experimental results and *K*_d_ values of different temperatures to calculate the thermodynamic parameters (∆*H*, ∆*S*, and ∆*G*) [25]:(8)$$\Delta G=-RT\ln K_{d}$$
(9)$$\ln K_{d}=\frac{\Delta S}{R}-\frac{\Delta H}{RT}$$
where *R* is the universal gas constant (8.314 J·mol^−1^·K^−1^). ln*K_d_* is plotted with 1/*T*, the slope and intercept are acquired, and ∆*H* and ∆*S* are calculated; and the thermodynamic parameters are listed in Table 2. ∆*G* values (−1.729 to −3.133 kJ·mol^−1^) were negative at various temperatures, proving that the process was feasible and spontaneous. In general, ∆*G* values between −20 and 0 kJ·mol^−1^ represent physisorption. The ∆*H* value (21.29 kJ·mol^−1^) indicated that the process was endothermic. During calcium ions adsorption on RMCRs, the positive value of ∆*S* indicated that the randomness between the solid/solution interfaces increased.

### 3.5. Effect of Remelt Syrup Brix

The calcium ions removal at different Brix values of the remelt syrup was investigated. As shown in Figure 6a, the removal rate of calcium ions decreased at 55 °Bx. Brix values of the remelt syrup decreasing in remelt syrup can decrease the mass concentration gradient pressure and viscosity. Brix values of the remelt syrup provide a power to overcome the bulky transfer resistance of calcium ions between solution and RMCRs [26]. Hence, the reduction in the initial remelt syrup Brix can enhance the interaction strength between calcium ions and RMCRs. The reduction in the initial remelted syrup sugar content can gain the interaction of calcium ions with the sugar content of the remelted syrup. Considering adsorption efficiency, we selected 45 °Bx of remelt syrup in the subsequent experiment. On the other hand, it also shows that RMCRs can adapt to the viscosity and pressure of syrup and is suitable for the sugar industry.

### 3.6. Comparison with Various Commercial Adsorbents

The removal rate of calcium ions on the RMCRs was compared with those on other commercial resins (i.e., FPA 51, FPA 98 Cl, FPC 22 Na, FPA 40 Cl, FPA 90 Cl, FPC 14 Na, FPC 23 H, and FPC 22 H) (Figure 6b). The main physicochemical properties of the commercial resins are presented in Table 3. The removal rates of calcium ions by FPA 51, FPA 98 Cl, FPC 22 Na, FPA 40 Cl, FPA 90 Cl, FPC 14 Na, FPC 23 H, FPC 22 H, and RMCR were 12.54%, 27.26%, 33.25%, 34.61%, 25.15%, 90.82%, 91.65%, 91.48%, and 90.71%, respectively. FPC 14Na, FPC23 H, FPC22 H, and RMCRs revealed the superior adsorption abilities for calcium ions, suggesting that they would be excellent adsorbents for removing calcium ions from remelt syrup.

Lead adsorption capacities vary depending on adsorbent properties such as structure, surface area, porosity, and adsorbent polarity. Compared with FPA 51, FPA 98 Cl, FPA 40 Cl, and FPA 90 Cl are all anion exchange resins, and FPC 14 Na, FPC 23 H, FPC 22 H, and RMCRs are all cation exchange resins with superior adsorption abilities for calcium ions, thereby showing that the polarity of the adsorbent is a key element determining the adsorption capacity. Therefore, RMCRs with carboxyl functional groups are a potential adsorbent in the sugar industry.

### 3.7. Regeneration

One of the important factors in evaluating adsorbent performance is reusability. After the adsorption, RMCRs were regenerated with HCl (0.1 mol·L^−1^) solutions, washed with deionized water until neutral, and used for the next adsorption experiments. The findings are shown in Figure 6c, and the regeneration efficiency was successively regenerated eight times. Even after eight regenerations, the RMCRs still contain a remarkable removal rate (80.87%). Hence RMCRs can be repeatedly used for the removal of calcium ions from remelt syrup.

### 3.8. Column Adsorption Performance and Models

Beyond the experiments already described, calcium ions from remelt syrup adsorption on RMCRs was estimated on fixed-bed columns. Figure 7 shows the relationship between *C*_t_/*C*_0_ and throughput volume when the bed depth is 4.0 cm. The breakthrough point is defined as the time when the effluent concentration reaches a percentage of the influent concentration (*C*_0_), which is considered unacceptable, e.g., 10% (*C*/*C*_0_ = 0:1). For *C*/*C*_0_ = 0.1, the number of bed volumes that pass through the adsorbent was 480 BV. The Thomas [27], Yoon–Nelson [28], and Adams–Bohart models [29] are devoted to estimate the sorption adsorption behavior.

The Thomas, Yoon–Nelson, and Adams–Bohart model equations can be written as follows, respectively:

Thomas model [27]:(10)$$\frac{C_{t}}{C_{0}}=\frac{1}{1+\exp(\frac{K_{Th}q_{0}m}{Q}-K_{Th}C_{0}t)}$$

Yoon–Nelson model [28]:(11)$$\frac{C_{t}}{C_{0}}=\frac{\exp(K_{YN}t-\tau K_{YN})}{1+\exp(K_{YN}t-\tau K_{YN})}$$

Adams–Bohart model [29]:(12)$$\frac{C_{t}}{C_{0}}=\exp(k_{AB}C_{0}t-k_{AB}N_{0}\frac{Z}{F})$$

In the above equations, *K_Th_*, *K_YN_*, and *k_AB_* are the Thomas rate constant (mL·min^−1^·mg^−1^), Yoon–Nelson rate constant (min^−1^), and the Adams–Bohart rate constant (min^−1^), respectively; *q*_0_ represents the column adsorption ability (mg·g^−1^), *Q* is the flow velocity (mL·min^−1^), and *m* is the mass of the RMCR (g). *C*_0_ and *C_t_* are the calcium ion concentrations at the inlet and outlet, respectively (mg·L^−1^). *τ* is the time (min) required for the adsorbate to breakthrough 50%. *t* (min) is the filtering time. *N*_0_ is the saturation concentration of the bed (mg·L^−1^), and *t_b_* is the service time at breakthrough (h).

From Table 4, the breakthrough is more fitted with the Thomas model (*R*^2^ = 0.966). According to Thomas model calculation, the dynamic saturated adsorption capacity of RMCRs for calcium ions from remelt syrup was 37.90 mg·g^−1^. The calculated value was close to the actual experimental results (37.05 mg·g^−1^). It was found that the Thomas model could be used to describe the dynamic adsorption characteristics of calcium ions adsorbed by RMCRs and predict the dynamic adsorption amount in industrial application.

### 3.9. Characterization of RMCRs before and after Adsorption of Calcium Ions

#### 3.9.1. FTIR Analysis

The FTIR spectra of RMCRs and RMCRs with adsorbed calcium ions are shown in Figure 8. The spectra show wide absorption peaks at 3410 and 3391.52 cm^−1^, which assign to the O−H bond stretching vibration in the hydroxyl function groups [30]. The bands at 2987 and 2937.93 cm^−1^ originate from the symmetry flex vibration of C−H bonds in −CH_2_−, which are derived from the concatenation of carbonaceous species in RMCRs and remelt syrup. Wavenumbers indicate the adsorption of organic ingredients deposited on the RMCRs [31]. After calcium ions adsorption, the asymmetric −CH_2_− stretching vibration shifts from 2987 cm^−1^ to 2937 cm^−1^, thus indicating interactions with alkyl chains of RMCRs. The bands of calcium ions adsorbed on RMCRs at 1716.37 cm^−1^ (C=O), 1558.22 cm^−1^ (COO−), and 911.26 cm^−1^ (C−O−C) are assigned to the C=O in ester carboxyl or carboxyl groups, which are attributed to the carboxyl group on RMCRs. This proved the interaction of calcium ions and the –COO– of RMCRs. The wavenumbers at 1457.5 cm^−1^ (C−OH), 1418.3 cm^−1^ (COO−), and 1345.25 cm^−1^ (C−N) are due to the existence of proteins in the remelt syrup [32,33,34,35]. The sucrose compounds and phenols characteristic bands include 1052.26 (C−O) and 3391.52 cm^−1^ (O−H). These peaks are due to polysaccharides from remelt syrup [36]. The above results demonstrate that polysaccharides, protein, phenols, and sucrose can also be adsorbed on RMCRs [36].

#### 3.9.2. XPS Analysis

XPS analysis was enforced to evaluate the elemental composites and chemical states RMCRs and RMCRs with adsorbed calcium ions, which are shown in Figure 8. The deconvolution of C 1s and O 1s peaks is also presented in Figure 8. The C 1s peaks of RMCRs (a) yield three contributions, which are 284.8 eV (C−C), 286.6 eV (C=O), and 288.1 eV (COO−) [37]. The shift in the carbon signal at 286.6 eV to lower BE after calcium ions adsorption is probably caused by the interaction of calcium ions with C−OH [38]. The peaks at 285.9 eV (C−C) and 287.4 eV (C=O) are related to sucrose compounds and phenols from remelt syrup. For the O 1s of RMCRs (b), the peak at 531.3 eV is attributable to C=O, that at 532.1 eV is attributable to C−O, and that at 535.8 eV is attributable to COO−. However, after calcium ions adsorption, the peak at 532.1 eV shifts to higher BE, which is caused by the interaction of calcium ions with oxygen atoms.

In Figure 8c, calcium ions are adsorbed through ionic bonding, thereby forming –(COO)_2_Ca. The peaks of Ca 2p_3/2_ at 347.2 and 346.9 eV represent the bonds between calcium ions and −COO−. The peak of Ca 2p_1/2_ at 350.6 eV is attributable to CaCO_3_ on the surfaces of the RMCRs. In Figure 8f, the Na 1s peak height of RMCRs with adsorbed calcium ions is lower than that of RMCRs; instead, the binding energy of Ca 2p at 346.6 eV is identified, which indicates that the ion exchange between −COONa and calcium ions contributes to calcium ions removal [39]. At the same time, it also shows that RMCRs have a certain ion exchange effect on the removal of calcium ions.

#### 3.9.3. EDS Analysis

The energy-dispersive X-ray spectroscopy (EDS) analysis (Figure 8g) of RMCRs suggests that the RMCRs contain carbon, oxygen, and a mass of sodium. EDS analysis (Figure 8h) implies that RMCRs with adsorbed calcium ions contains a spot of sodium, calcium, and potassium. Carbon, oxygen, and sodium come from RMCRs; potassium and calcium are from remelt syrup. The EDS spectra of RMCRs and RMCRs with adsorbed calcium ions suggest that the ion exchange may drive the uptake process of calcium ions.

The comprehensive analysis of FTIR, XPS, and EDS showed that the calcium ions from remelt syrup were adsorbed on RMCRs in this work. RMCRs have a superior adsorption effect on calcium ions and have great potential for application in the sugar industry.

## 4. Conclusions

This work investigates the potential of RMCRs for calcium ions removal from remelt syrup. The results show that the maximum monolayer adsorption capacity of calcium ions is 37.05 mg·g^−1^ at a resin dosage of 4 g·L^−1^, pH of 7.0, temperature of 75 °C, and contact time of 10 h. The removal rate of calcium ions from remelt syrup by RMCRs is 90.71%. The adsorption of calcium ions on RMCRs is pseudo-first-order in proportion and conforms to the Freundlich isotherm model. The adsorption process is endothermic, the adsorption process is physical adsorption and involves weak chemical bonds, and the analyses of FTIR, XPS, and EDS prove that ion exchange occurs during the adsorption process. The Thomas model describes the dynamic adsorption well. Compared with commercial resins, RMCRs have a superior removal rate for calcium ions from remelt syrup. In summary, RMCRs can be used as adsorbents for removal of calcium ions from remelt syrup, and potentially useful in improving the quality of remelt syrup and reducing or eliminating the use of chemicals in the sugar industry.

## Figures and Tables

**Figure 1 polymers-14-02397-f001:**
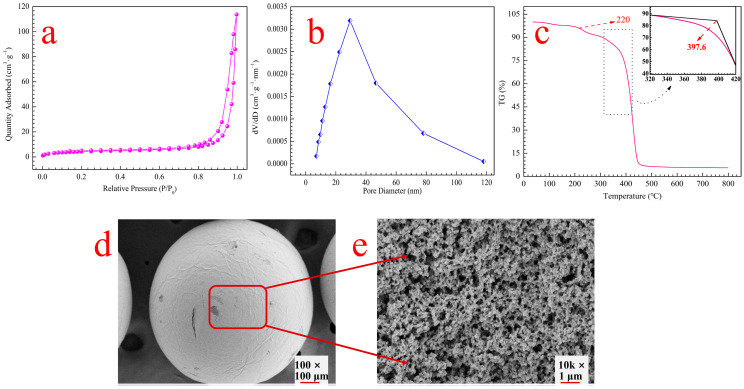
Characterization of the RMCRs by N_2_ adsorption–desorption isotherms (**a**), pore size distributions (**b**), TGA (**c**), and SEM (**d**,**e**).

**Figure 2 polymers-14-02397-f002:**
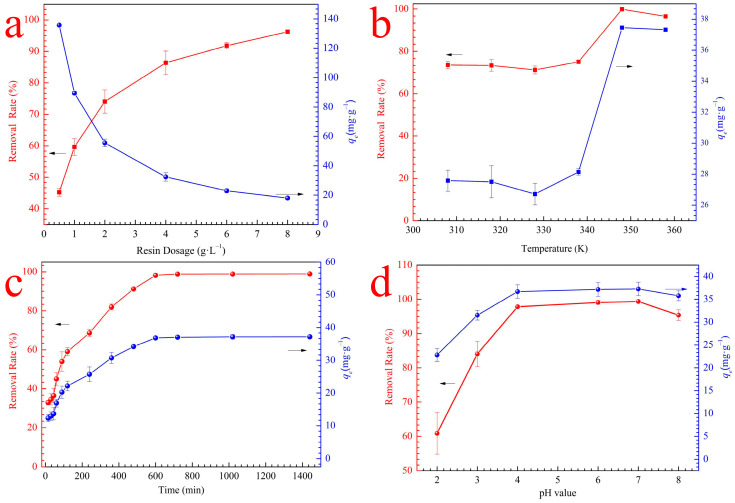
The adsorption effect of calcium ions adsorbed by RMCRs as a function of adsorbent dosage (**a**), temperature (**b**), contact time (**c**), and pH value of RMCRs (**d**).

**Figure 3 polymers-14-02397-f003:**
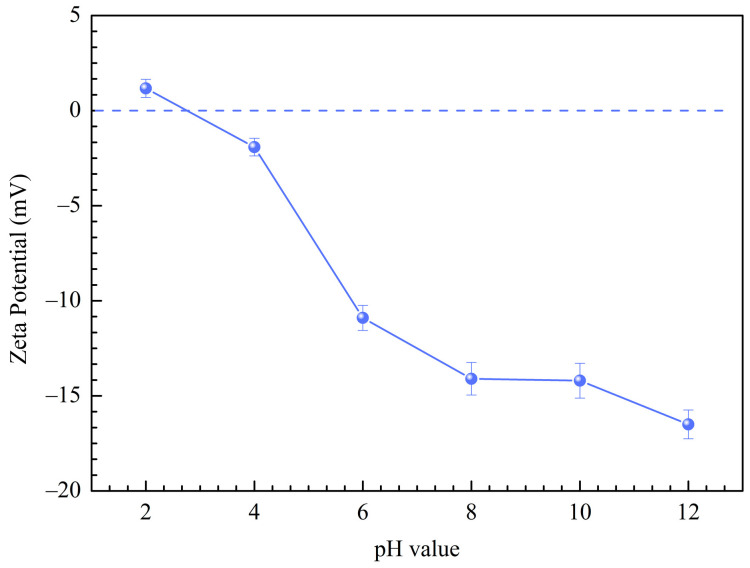
Zeta potentials of RMCRs at different pH.

**Figure 4 polymers-14-02397-f004:**
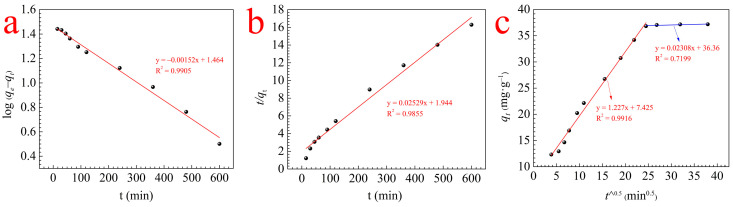
Behavior of RMCR adsorption of calcium ions: fits of the kinetic data of calcium ions adsorption on RMCRs using pseudo-first-order (**a**), pseudo-second-order (**b**), and intraparticle diffusion models (**c**).

**Figure 5 polymers-14-02397-f005:**
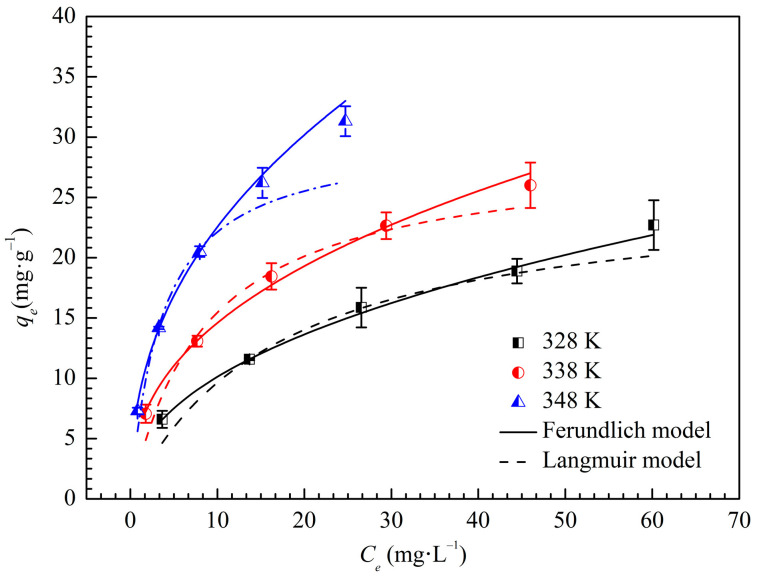
Isotherms for the adsorption of calcium ions on RMCRs at 328, 338, and 348 K.

**Figure 6 polymers-14-02397-f006:**
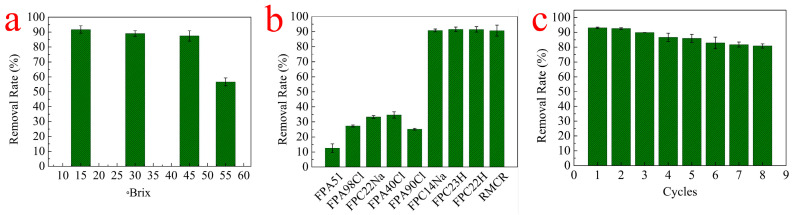
Effect of remelt syrup Brix on calcium ions adsorption onto RMCRs (**a**), comparison with various commercial resins (**b**), and reusability of RMCRs for calcium ions uptake (**c**).

**Figure 7 polymers-14-02397-f007:**
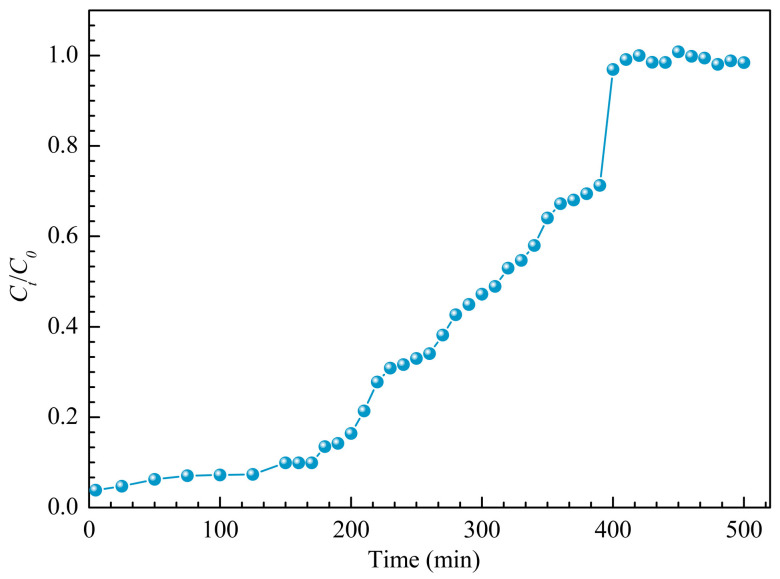
Breakthrough curve for calcium ions adsorption by RMCRs fixed columns.

**Figure 8 polymers-14-02397-f008:**
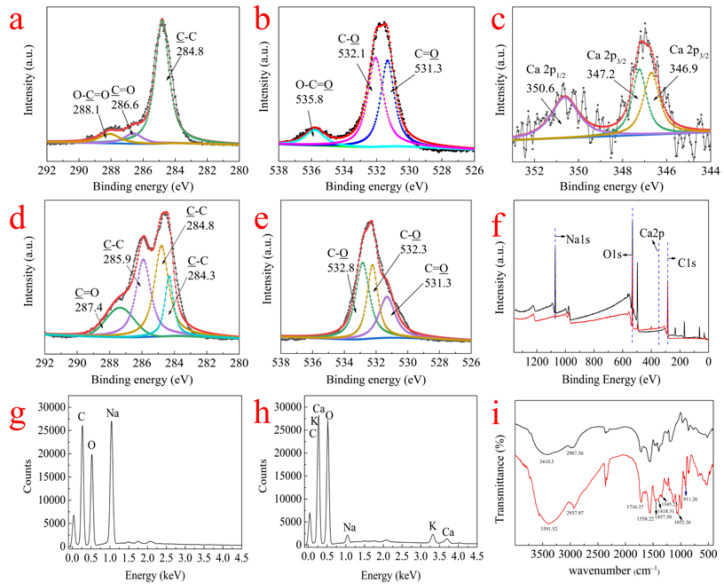
XPS C1s (**a**,**d**) and O1s (**b**,**e**) spectra of RMCRs and their deconvolution into component peaks before and after adsorption of calcium ions, respectively, and XPS spectra of calcium ions (**c**) for RMCRs with adsorbed calcium ions, XPS fully scanned spectra (**f**), EDS spectra of RMCRs (**g**) and RMCRs with adsorbed calcium ions (**h**), and FTIR spectra (**i**) of RMCRs (black solid line) and RMCRs with adsorbed calcium ions (red solid line).

**Table 1 polymers-14-02397-t001:** Adsorption isotherm parameters of calcium ions onto RMCRs.

Temperature (*K*)	Freundlich Constants			Langmuir Constants		
	1/*n*	*K_F_* (mg·g^−1^)	*R* ^2^	*q_m_* (mg·g^−1^)	*K_L_* (L·mg^−1^)	*R* ^2^
328	0.4353	3.586	0.9974	25.75	0.05989	0.9088
338	0.4056	4.735	0.9957	28.79	0.1158	0.9313
348	0.4280	5.716	0.9954	30.13	0.2768	0.9282

**Table 2 polymers-14-02397-t002:** Thermodynamic parameters of the calcium ions adsorption onto RMCRs.

∆*H* (kJ·mol^−1^)	∆*S* (J·mol^−1^·K^−1^)	∆*G* (kJ·mol^−1^)		
		328 K	338 K	348 K
21.29	70.18	−1.729	−2.431	−3.133

**Table 3 polymers-14-02397-t003:** Physicochemical properties of the commercial resins used.

Commercial Resins	Particle Size (mm)	Exchange Capacity (eq·L^−1^)	Matrix Structure	Functional Group
FPA51	0.49–0.69	≥1.3	Styrene-divinylbenzene copolymer	−NR_2_
FPA98 Cl	0.63–0.85	≥0.8	polymethacrylic acid	R_4_NOH
FPC22 Na	0.60–0.80	≥1.7	Styrene-divinylbenzene copolymer	−SO_3_Na
FPA40 Cl	0.50–0.75	≥1.0	Styrene-divinylbenzene copolymer	R_4_NOH
FPA90 Cl	0.65–0.82	≥1.0	Styrene-divinylbenzene copolymer	R_4_NOH
FPC14 Na	0.60–0.80	≥2.0	Styrene-divinylbenzene copolymer	−SO_3_Na
FPC23 H	0.58–0.80	≥2.2	Styrene-divinylbenzene copolymer	−SO_3_H
FPC22 H	0.60–0.80	≥1.7	Styrene-divinylbenzene copolymer	−SO_3_H
RMCR	0.35–0.83	≥0.3	polymethacrylic acid	−COONa

**Table 4 polymers-14-02397-t004:** Model parameters for RMCR fixed-bed columns calcium ions.

Thomas Model			Yoon-Nelson Model			Adams-Bohart Model			
*q*_0_ (mg·g^−1^)	*K_Th_* × 10^−5^ (L·mg^−1^·min^−1^)	*R* ^2^	*K_YN_* × 10^−2^ (min^−1^)	*τ* (min)	*R* ^2^	*Z* (cm)	*N*_0_ (mg·L^−1^)	*k_AB_* × 10^−5^ (L·mg^−1^·min^−1^)	*R* ^2^
37.90	6.440	0.9666	1.165	314.3	0.9666	4	484313	4.616	0.9638

## Data Availability

The data presented in this study are available on request from the corresponding author.

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
