# Peer review of "Removing Calcium Ions from Remelt Syrup with Rosin-Based Macroporous Cationic Resin"

_polymers, 2022, doi:10.3390/polym14122397_

Round 1
Reviewer 1 Report
Authors should address several point, and the main is to stress what is the importance of their work for polymer science and technology. In particular authors should improve:
Use of R2 for evaluating the quality of experimental model is not a good approach. This parameter only evaluates the average error of the fitting. However, It can assume very high values by cancellation of individual errors with high value. Therefore, any comment based on this parameter should be removed from the text.
The impact for polymer technology and science is not clearly stated, and it is essential
There are no significant discussion of the data. There are many results. However, there is a lacked discussion.
English and grammar should be revised within the manuscript.
Author Response
Comments:
Authors should address several point, and the main is to stress what is the importance of their work for polymer science and technology.
Point 1: What is the importance of their work for polymer science and technology.
Response 1: This paper introduces the synthesis of rosin-based macroporous cationic resins (RMCRs) and the application of removing calcium ions from remelt syrup, which is closely related to polymer science and technology. The RMCRs used in this study were all self-prepared by suspension polymerization and followed a previous preparation process with some modifications. The functional monomers MAA (6.32 g), the porogen polypropylene glycol (1.39 g), the cross-linker EGMEA (20.25 g), and AIBN (0.2 g) were dissolved in ethyl acetate (60 mL) by sonication to obtain an organic phase. SDS (0.02 g) and PVA (0.02 g) were dissolved in deionized water in a 250 mL three-necked flask, and then the organic phase was added at 60 °C. The mixture was thermally polymerized at 80 °C for 8 h and stirred at 200 rpm. The resins were extracted with ethanol and deionized water, then immersed in 3.0% NaOH solution to ionize the COOH groups to COO− groups, and ultimately washed continuously with deionized water until the pH was approximately 7.0. Moreover, the description regarding to this issue has been added in the revised manuscript.
Point 2: Use of R2 for evaluating the quality of experimental model is not a good approach. This parameter only evaluates the average error of the fitting. However, it can assume very high values by cancellation of individual errors with high value. Therefore, any comment based on this parameter should be removed from the text.
Response 2: Many thanks for your suggestion. The use of R2 for evaluating the quality of experimental model is not a good approach. This parameter only evaluates the average error of the fitting. It can assume very high values by the cancellation of individual errors with high values. So we can combine other parameters and correlation coefficients in the model to fit and explain which model may be more inclined to. In the revised manuscript, we have supplemented the description of other parameters in the model.
Point 3: There are no significant discussion of the data. There are many results. However, there is a lacked discussion.
Response 3: Many thanks for this suggestion. We have supplemented the discussion of the data in the revised manuscript.
Point 4: English and grammar should be revised within the manuscript.
Response 4: Sorry for our carelessness. We found some grammatical errors through re-examination, and we have made corresponding changes.
Reviewer 2 Report
This paper describes the calcium ions adsorption performance of rosin-based macroporous cationic resins in remelt syrup. The manuscript comprises an interesting introduction, followed by a detailed description of the preparation and investigation methods, which are coherently supported by the reported data. The authors conclude that the RMCRs have potential applications as inexpensive and efficient adsorbents for calcium ions.
The paper is well organized and the results are interesting and suitable for publication. However, in order to improve the manuscript, I strongly recommend comparing their results with the existing references and reporting their new insights in this field.
Author Response
Comments:
This paper describes the calcium ions adsorption performance of rosin-based macroporous cationic resins in remelt syrup. The manuscript comprises an interesting introduction, followed by a detailed description of the preparation and investigation methods, which are coherently supported by the reported data. The authors conclude that the RMCRs have potential applications as inexpensive and efficient adsorbents for calcium ions.
The paper is well organized and the results are interesting and suitable for publication.
Point 1: However, in order to improve the manuscript, I strongly recommend comparing their results with the existing references and reporting their new insights in this field.
Response 1: Thanks for your suggestion. We reviewed the relevant literature and supplemented it in the revised manuscript. Helmut and Joachim [1] claimed that calcium ions in beet juice evaporator could be reduced by 80–90% using KEBO DS(a scale inhibitor). Typically, anionic polymers, such as polyacrylics and poly (amino polyether tetra-methylene phosphonic acid), are used in the sugar industry to inhibit calcium salts [2]. The use of scale inhibitors in the sugar industry is also limited by health concerns. These inhibitors need to be approved by relevant agencies (for example, the United States Food and Drug Administration) before they can be used in the sugar process [2]. Future research directions include the development of multifunctional nanoparticles and green biomass-based adsorbents to significantly improve impurity removal in the sugar industry [3]. RMCRs were polymerized by rosin‐based crosslinker EGMRA, which is green, economical, eco‐friendly, and more suitable for the sugar industry. Furthermore, the removal rate of calcium ions on the RMCRs is compared with that on other commercial resins (Figure 6b). RMCRs showed good superiority in removing calcium ions from remelt syrup.
[1] Helmut, B.; Joachim, P.H., Scales in evaporators. International Sugar Journal 2003, 105, 475-480.
[2] East, C.P.; Fellows, C.M. Doherty, W.O.S., Chapter 25‐Scale in Sugar Juice Evaporators: Types, Cases, and Prevention, in Mineral Scales and Deposits. 2015. pp. 619-637.
[3] Bakir, C.H.; Rackemann, D.; Doherty, W., Current perspective and future research directions on defecation clarification for the manufacture of raw sugar. International Sugar Journal 2021, 146, 634-642.
Reviewer 3 Report
The manuscript needs some major revision before acceptance:
RMCRS was previously prepared (Reference 16). What is the novelty of the current work?
Figure 1 (a, b) are not clear. redraw them to be more readable.
Magnification of the two SEM images should be added to the images.
What is the facts that illustrate from zeta potential figure?
Author Response
Comments:
The manuscript needs some major revision before acceptance:
Point 1: RMCRS was previously prepared (Reference 16). What is the novelty of the current work?
Response 1: The RMCRs used in this study were all self-prepared and followed a previous preparation process with some modifications. In order to improve the adsorption capacity of the RMCRs, the proportion of porogen and monomer was adjusted to make it have a larger surface area and more abundant COOH groups. This study is the first to apply the use of the RMCRs in removing calcium ions from remelt syrup. Furthermore, the mechanisms of the RMCRs adsorption of the calcium ions were elucidated by using adsorptive isotherm and kinetic models and calculating thermodynamic parameters. This study provides a theoretical basis for the application of RMCRs in sugar industry.
Point 2: Figure 1 (a, b) are not clear. redraw them to be more readable.
Response 2: Thanks for your suggestion. We have redrawn Figure 1 (a, b) in the revised manuscript.
Point 3: Magnification of the two SEM images should be added to the images.
Response 3: Many thanks for this suggestion. We have added magnification of the two SEM images in the revised manuscript.
Point 4: What is the facts that illustrate from zeta potential figure?
Response 4: The zeta potential value is correlated with the charge of the RMCRs and reflects their adsorption characteristic. Here, we focused on the main functional group of RMCRs involved in calcium ions adsorption, namely, the −COONa groups. This group could be very easily ionized to form −COO− groups. The zeta potentials of RMCRs are shown in Figure 3, with pHpzc (zero charge points) values of 2.7. As shown in Figure 2d, the removal percentages of calcium ions onto RMCRs increase as the pH increase. At pH 2.0, which is below pHpzc, the surfaces of RMCRs are positively charged, and so the adsorption of calcium ions is interrupted by electrostatic repulsion. When the solution pH > pHpzc, the surfaces of the RMCRs acquire negative charges. In the range of pH 2.0 to 3.0, the removal percentages rise from 60.87% to 84.08%. Thus, the calcium ions onto RMCRs must have been based on electrostatic attraction.
Round 2
Reviewer 1 Report
Now the article is publishable
Reviewer 3 Report
Authors address all comments